# Predictors for participation in mass-treatment and female genital schistosomiasis re-investigation, and the effect of praziquantel treatment in South African adolescents

**Takalani Girly Nemungadi**[1]*, **Elisabeth Kleppa**[2], **Hashini Nilushika Galappaththi-Arachchige**[2], **Pavitra Pillay**[3], **Svein Gunnar Gundersen**[4], **Birgitte Jyding Vennervald**[5], **Patricia Doris Ndhlovu**[6], **Myra Taylor**[1], **Saloshni Naidoo**[1], **Eyrun Floerecke Kjetland**[1,2]

1 Discipline of Public Health Medicine, College of Health Sciences, University of KwaZulu-Natal, Durban, South Africa, 2 Norwegian Centre for Imported and Tropical Diseases, Oslo University Hospital, Oslo, Norway, 3 Department of Biomedical and Clinical Technology, Durban University of Technology, KwaZulu-Natal, Durban, South Africa, 4 Institute for Global Development and Planning, University of Agder, Kristiansand, Norway, 5 Section for Parasitology and Aquatic Pathobiology, Faculty of Health and Medical Sciences, University of Copenhagen, Copenhagen, Denmark, 6 BRIGHT Academy, Ugu District, South Africa

* takalaninemungadi@gmail.com

**Data Availability Statement:** Data cannot be shared publicly because of confidentiality. Data are

## Abstract

### Objective

Female Genital Schistosomiasis (FGS) causes intravaginal lesions and symptoms that could be mistaken for sexually transmitted diseases or cancer. In adults, FGS lesions [grainy sandy patches (GSP), homogenous yellow patches (HYP), abnormal blood vessels and rubbery papules] are refractory to treatment. The effect of treatment has never been explored in young women; it is unclear if gynaecological investigation will be possible in this young age group (16–23 years). We explored the predictors for accepting anti-schistosomal treatment and/or gynaecological reinvestigation in young women, and the effects of anti-schistosomal mass-treatment (praziquantel) on the clinical manifestations of FGS at an adolescent age.

### Method

The study was conducted between 2011 and 2013 in randomly selected, rural, high schools in Ilembe, uThungulu and Ugu Districts, KwaZulu-Natal Province, East Coast of South Africa. At baseline, gynaecological investigations were conducted in female learners in grades 8 to 12, aged 16–23 years (n = 2293). Mass-treatment was offered in the low-transmission season between May and August (a few in September, n = 48), in accordance with WHO recommendations. Reinvestigation was offered after a median of 9 months (range 5–14 months). Univariate, multivariable and logistic regression analysis were used to measure the association between variables.

available from the Centre for Bilharzia and Tropical Health Research for researchers who meet the criteria for access to confidential data. Email address for request: brightresearch.cbthr@gmail.com.

**Funding:** This work was supported by the University of KwaZulu-Natal College of Health Sciences PhD Scholarship (student number 216073797). The research leading to these results has received funding from the European Research Council under the European Union's Seventh Framework Programme (FP7/2007-2013)/ERC Grant agreement no. PIRSES-GA-2010-269245, University of Copenhagen with the support from the Bill and Melinda Gates Foundation, Grant # OPPGH5344, and South-Eastern Regional Health Authority, Norway project no. 2016055. The funders had no role in study design, data collection and analysis, decision to publish, or preparation of the manuscript.

**Competing interests:** The authors have declared that no competing interests exist.

## Results

Prevalence: Of the 2293 learners who came for baseline gynaecological investigations, 1045 (46%) had FGS lesions and/or schistosomiasis, 209/1045 (20%) had GSP; 208/1045 (20%) HYP; 772/1045 (74%) had abnormal blood vessels; and 404/1045 (39%) were urine positive.

Overall participation rate for mass treatment and gynaecological investigation: Only 26% (587/2293) learners participated in the mass treatment and 17% (401/2293) participated in the follow up gynaecological reinvestigations.

Loss to follow-up among those with FGS: More than 70% of learners with FGS lesions at baseline were lost to follow-up for gynaecological investigations: 156/209 (75%) GSP; 154/208 (74%) HYP; 539/722 (75%) abnormal blood vessels; 238/404 (59%) urine positive. The grade 12 pupil had left school and did not participate in the reinvestigations (n = 375; 16%).

Follow-up findings: Amongst those with lesions who came for both treatment and reinvestigation, 12/19 still had GSP, 8/28 had HYP, and 54/90 had abnormal blood vessels. Only 3/55 remained positive for *S. haematobium* ova.

Factors influencing treatment and follow-up gynaecological investigation: HIV, current water contact, water contact as a toddler and urinary schistosomiasis influenced participation in mass treatment. Grainy sandy patches, abnormal blood vessels, HYP, previous pregnancy, current water contact, water contact as a toddler and father present in the family were strongly associated with coming back for follow-up gynaecological investigation.

Challenges in sample size for follow-up analysis of the effect of treatment: The low mass treatment uptake and loss to follow up among those who had baseline FGS reduced the chances of a larger sample size at follow up investigation. However, multivariable analysis showed that treatment had effect on the abnormal blood vessels (adjusted odds ratio = 2.1, 95% CI 1.1–3.9 and p = 0.018).

## Conclusion

Compliance to treatment and gynaecological reinvestigation was very low. There is need to embark on large scale awareness and advocacy in schools and communities before implementing mass-treatment and investigation studies. Despite challenges in sample size and significant loss to follow-up, limiting the ability to fully understand the treatment's effect, multivariable analysis demonstrated a significant treatment effect on abnormal blood vessels.

### Author summary

Female genital schistosomiasis (FGS) is a neglected tropical disease and it affects many women and young girls in schistosomiasis endemic areas. A lot of research is still needed to understand the characteristics of FGS, its prevention, as well as the timing for treatment. As a result of the limited information, some women who suffer from FGS end up being misdiagnosed with diseases such as human papilloma virus or other sexually transmitted diseases. The study highlights issues that need to be taken into considerations when providing treatment or conducting mass treatment for schistosomiasis and FGS or planning gynaecological investigations to inform FGS control programmes. In this study of adolescent girls and young women of KwaZulu-Natal Province of South Africa, we

sought to explore the factors that influence participation in mass treatment and gynaecological investigation, and investigating the effect of praziquantel treatment on FGS. Factors that influenced participation in mass treatment included HIV, current water contact, water contact as a toddler and urinary schistosomiasis. Factors that influenced participation in follow up gynaecological investigation included grainy sandy patches, abnormal blood vessels, homogenous yellow patches, previous pregnancy, current water contact, water contact as a toddler and father present in the family. There was low uptake and loss to follow up for mass treatment, and this contributed to small sample size for follow up gynaecological investigations to understand the effect of treatment. However, multivariable analysis showed that treatment had effect on the abnormal blood vessels and not on grainy sandy patches and homogenous yellow patches.

## Introduction

Female Genital Schistosomiasis (FGS) is a complication of schistosomiasis that results from trapped schistosome eggs, which damage tissues and organs [1,2]. It is known to affect females of all age groups. FGS has a complex disease manifestation spectrum, including lesions of the cervix and vagina (grainy sandy patches and homogenous yellow patches), surface bleeding, abnormal blood vessels and rubbery papules [3,4]. FGS has also been associated with sub-fertility or infertility, ectopic pregnancy, spontaneous abortion, premature birth, and increased susceptibility to HIV transmission and possibly progression. Reports suggest that eggs are evenly distributed in the genital organs [2,5–7]. There are many case reports of concurrent FGS with cervical intra-epithelial neoplasia and one paper to date has shown lower Human Papillomavirus (HPV) clearance in FGS positive women or increased susceptibility to HPV [3,8–12].

WHO recommends mass drug administration in schistosomiasis endemic schools for prevention of morbidity, and studies indicate that early treatment may be necessary to prevent genital lesions in adulthood [13–15]. **In South Africa, there is no mass treatment or mass drug administration programme** for schistosomiasis and treatment is case-by-case upon consultation by the affected individuals. The mass treatment that was conducted between 1998 and 2001 in KwaZulu-Natal was discontinued due to resource constraints [16,17]. FGS is not known among most health care workers, clinicians and community members and only approximately 160 gynaecologists were previously informed about FGS in South Africa [16–19]. As a result, FGS is not considered a priority condition. There is therefore limited data and research and this leads to a low index of suspicion by clinicians and FGS lesions often being misclassified as sexually transmitted infections or cervical cancer [2,5,20–27]. There have been efforts by authorities in South Africa to mobilise WHO-donated praziquantel in order to implement the mass treatment as per the recommended WHO strategy over the past years. However, this has been a challenge since, the donated praziquantel is not registered in the country and is not permitted to be used [28]. Subsequently, treatment has been mobilised from within the country and mass treatment is being planned for implementation [29]. In order to prevent low uptake and to properly plan for the roll-out of the mass treatment, it is important to identify factors that will influence participation, and to determine the effect of treatment on FGS symptoms. **It is also important to identify factors that will influence participation in gynaecological investigation in order to improve treatment and other prevention and control measures as well as research information that will provide new knowledge.**

Between 2011 and 2013, we conducted a follow-up study and sought to explore the feasibility of the implementation of mass drug administration and gynaecological investigations in school-going young women in rural KwaZulu-Natal, and to explore the effects on lesions and symptoms. This is in line with the WHO recommendations of promoting regular de-worming and expanding the reach of at risk adolescent girls and women of reproductive age [14].

## Materials and methods

### Ethics statement

The study was approved by the Biomedical Research Ethics Committee (BREC), University of KwaZulu-Natal (Ref BF029/07), KwaZulu-Natal Department of Health (Reference HRKM010-08) and the Regional Committee for Medical and Health Research Ethics (REC), South Eastern Norway. The ethical committees were aware that minors were invited into the study and specifically approved independent minor consent without parental consent. All study participants were offered anti-schistosomal treatment and, if applicable, treatment for sexually transmitted diseases, and/or referral to the local health system for treatment of HIV.

### Participants and area

The study was conducted between 2011 and 2013 in the KwaZulu-Natal Province of South Africa. The participants were female learners aged 16–23 years, from randomly selected high schools in Ilembe, uThungulu and Ugu Districts on the East Coast of South Africa that had not undergone anti-schistosomal mass-treatment before the first investigation as shown in Table 1. The participants were recruited from schools that were classified as rural by the South African Department of Basic Education and were below the altitude of 400 meters above sea level, with an estimated prevalence of *S. haematobium* of 10% or more based on an initial show of hands for red urine in Ugu District and a haematuria dipstick survey in Ilembe and uThungulu districts [30]. Gynaecological examinations were performed in two research clinics (North of Durban and South of Durban); virgins, pregnant, and severely ill females were excluded. Schools with prevalence of less than 10% were excluded.

### Questionnaires and clinical examinations

Group information, individual information and the consent form procedure was done at the school over a 2-week-period. Consent forms and parent information were distributed in advance [39–41]. The investigation has been described previously. Briefly, a questionnaire on water contact, reproductive history, genital and abdominal symptoms was administered individually to participants in isiZulu (the local language) prior to gynaecological examination [30]. Teachers would suggest days when it would be suitable to fetch learners from schools to go to the research clinic. Teenagers had indicated they were embarrassed to give urine in school so urine was collected only while they were at the clinic. On pre-reserved days, a female driver, trained and well-informed about the study, would pick up 4–13 participants (depending on the vehicle size) for the drive to the clinic for gynaecological examinations by trained medical doctors to determine the presence or absence of FGS lesions using a colposcopy. For quality control, the computer images analysis was done afterwards by an FGS specialist as the supervisor who has worked in five countries since 1994.

### Treatment

In accordance with the WHO policy for areas of high schistosomiasis endemicity, mass treatment of participants and other learners in enrolled schools was carried out during the winter

**Table 1. Study design and practicalities, their rationale, and the potential unintended consequences.**

| Inclusion considerations | Reason |
|---|---|
| Participants included were 16 years and above, including Grade 12 | In order to explore reversibility of lesions, we wished to invite young women at an earliest possible time for a gynaecological investigation. However, gynaecological investigations are not practically feasible before the age of 16 for cultural, legal and sometimes anatomical reasons [31]. Average sexual debut age is 18.7 years in South Africa and many hide sexual debut from e.g. their parents and teachers for as long as they can [32,33]. Moreover, some may be worried about the "loss of virginity" during speculum examination [34,35]. Visual inspection of the cervix, fornices and vaginal surfaces before sexual debut may therefore be offensive to their sensibilities. |
| KwaZulu-Natal was chosen as the study area | Firstly, at this tender age, chances were high that these young women would be invited for their first gynaecological examination. Some amongst the pupils might be found to have been raped and a gynaecological examination might trigger psychological reactions. Therefore, we looked for a study area where psychologists and support centres would be available for study participants in a schistosomiasis endemic area. Secondly, there are more than 4,000 rural schools in KZN and we estimated that we would be able to reach the necessary sample size indicated below in the sample calculation under material and method [36,37]. |
| Although some young women with multiple lesions were treated at the clinic, most were not treated immediately after examination at the clinic. | Treatment with praziquantel was done at schools, not at the research clinics for the following reasons: 1) Mass Drug Administration is recommended so that siblings and class mates are treated simultaneously [15] 2) Anti-schistosomal treatment with praziquantel should be given in the low-transmission season [28,38] 3) Praziquantel may cause nausea and sometimes vomiting; if treatment had been done at clinic immediately after investigation, some learners might become car sick on the way from the clinic, it could potentially scare others from participating [15]. |

season (low-transmission season) after the baseline gynaecological examination. Treatment began after lunch to ensure that the children had eaten their free school lunches before treatment and to save the costs on food for the programme. Depending on the size of school, the treatment team comprised of 2–4 nurses and 2–4 assistants [27]. Bread and bananas were distributed to learners who had not received a school lunch. Learners with signed consent forms were weighed, and the number of praziquantel tablets was calculated for the dose of 40 mg per kg of body weight. As described previously, a designated nurse directly observed all tablet ingestion by counting the number of tablets in each learner's hand and watching for hand-to-mouth intake [27]. The treatment team stayed in the school for an hour after the last dose to allow learners to report any side effects. All learners, even the untreated, were allowed to come back for follow-up gynaecological examination.

## Sample size calculation and statistical analyses

Demographic and clinical variables were explored as predictors for (1) participation in mass-treatment and (2) returning to a follow-up gynaecological investigation. We planned a study of independent cases and controls with 3 control(s) per case. Downs et al described that only 2 out of 9 women with FGS lesions were healed 6 months months/weeks after treatment with

praziquantel [42]. However, there was no control group. There is only one paper that has explored the effect of treatment (3 and 12 months) on the FGS lesions and compared it with the untreated [43]. We planned to study independent treated and untreated FGS positive females. In this young population, where lesions might possibly be reversible, the authors decided to explore 50% reversibility of the FGS lesions in the treated. This is low, a very conservative estimate, due to the lack of prior data. We therefore estimated the probability of healing among the treated FGS positive to be 0.5. Amongst the untreated, we assumed that the probability of healing would be 0.1. We would therefore need to study 19 FGS positive patients and 19 FGS negative patients to be able to reject the null hypothesis that the exposure rates for case and controls are equal with probability (power) 0.8. The Type I error probability associated with this test of this null hypothesis is 0.05. We used an uncorrected chi-squared statistic to evaluate this null hypothesis.

In order to study simultaneously the impact of several variables, logistic regression analysis was applied with a 5% significance level; as a general rule variables were included if the p-value from the crude association was less than 0.2 and if the Spearman rank correlation coefficient was below 0.7. Age was forced into the model.

To assess the overlap of the FGS lesions among the learners, we created Venn diagrams (presented under results) using Venny version 2.1 (Juan Carlos Oliveros, https://bioinfogp.cnb.csic.es/tools/venny/index2.0.2.html)

## Results

### Description of the study population

A total of 2293 adolescent girls and young women from 70 secondary schools were enrolled and examined gynaecologically for FGS lesions and urinary schistosomiasis at baseline. As shown in Fig 1, 1045/2293 (46%) learners had genital lesions and/or urinary schistosomiasis at baseline. None of them had rubbery papules. Abnormal blood vessels were the most common findings.

As shown in Fig 2, 271 learners had more than one FGS lesions while 556 had only one FGS lesion (ABV: abnormal blood vessels, GSP: grainy sandy patches, HYP: homogenous yellow patches).

### Mass treatment participation

Treatment with praziquantel (40mg/kg) was given in the low-transmission season [South African winter between May and August, including a few in September (n = 48)]. Only 26% (587 of the 2293) learners were treated between baseline and follow-up. Multivariable analysis showed that HIV, current water contact, water contact as a toddler and urinary schistosomiasis positively influenced learners' willingness to participate in the mass treatment (Table 2). A

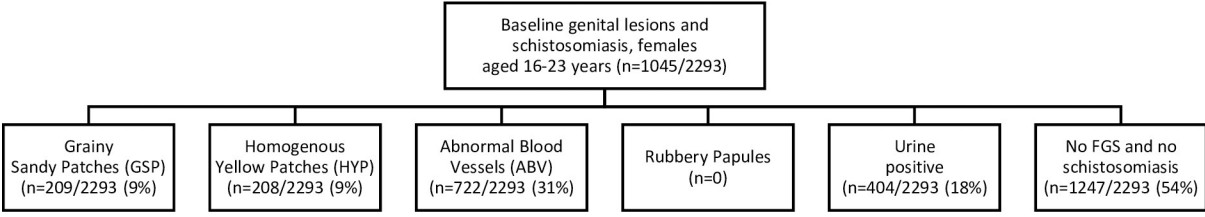

**Fig 1. Genital lesions and schistosomiasis amongst adolescent girls and young women in Kwa-Zulu-Natal, 2012–2013.**

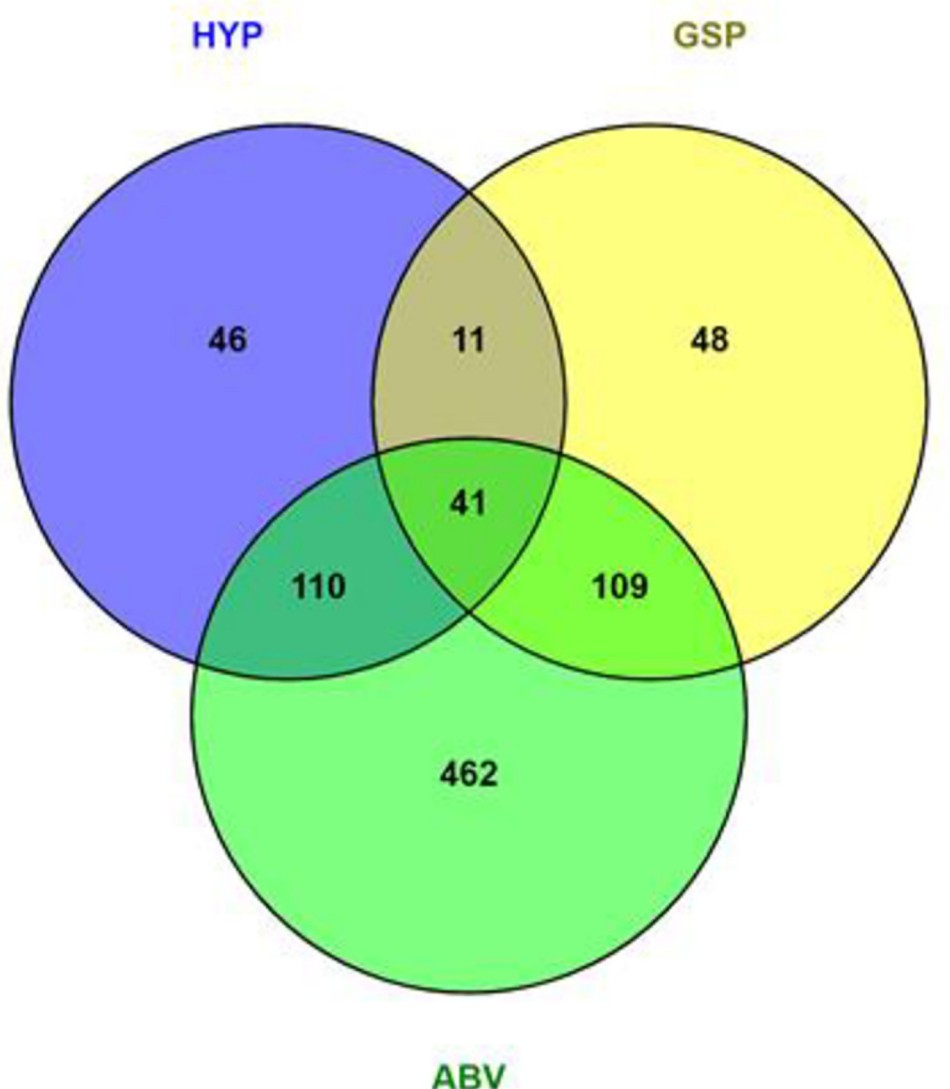

**Fig 2. Venn diagram showing an overlap of baseline FGS lesions among learners.**

greater loss to follow-up for mass treatment was observed in the Southern research clinic as compared to the Northern clinic (odds ratio = 2.1, 95% CI 1.7–2.6 and p <0.001).

## Gynaecological reinvestigation

Table 3 shows that a total of 709 learners (31% of the 2293 learners) were interviewed at follow up. Of the 709, 398 agreed to participate in the follow up gynaecological investigation; the majority (32%) felt that they were not ready. The willing participants were reinvestigated after a median of 9 months (range 5–14 months). The median age of those followed-up was 19 years (range 16–23 years), and those who were lost to follow-up had a median age of 18 years (range 16–23 years).

Table 4 show factors that may have contributed to compliance with follow up gynaecological investigation. The three FGS lesions (grainy sandy patches, abnormal blood vessels and

**Table 2. Underlying factors and compliance with mass-treatment.**

| History | Participated in mass treatment (%) | Did not participate in mass treatment (%) | Univariate analysis | | | *Multivariable analysis | | |
|---|---|---|---|---|---|---|---|---|
| | | | Odds ratio | 95% Confidence Interval | p-value | Adjusted odds ratio | 95% Confidence Interval | p-value |
| Grade 8 | 8/585 (1) | 10/1046 (1) | 1 | | | 1 | | |
| Grade 9 | 34/585 (6) | 45/1046 (4) | 0.9 | 0.3–2.6 | 0.913 | 0.7 | 0.2–2.3 | 0.600 |
| Grade 10 | 135/585 (23) | 287/1046 (27) | 0.6 | 0.2–1.5 | 0.274 | 0.4 | 0.2–1.2 | 0.113 |
| Grade 11 | 244/585 (42) | 474/1046 (45) | 0.6 | 0.3–1.7 | 0.389 | 0.5 | 0.2–1.5 | 0.249 |
| Grade 12 | 164/585 (28) | 230/1046 (22) | 0.9 | 0.3–2.3 | 0.813 | 0.6 | 0.2–0.7 | 0.346 |
| Current water contact | 391/587 (59) | 596/1050 (57) | 1.5 | 1.2–1.9 | <0.001 | 1.4 | 0.1–1.8 | 0.003 |
| Urinary schistosomiasis | 90/517 (17) | 193/895 (22) | 0.8 | 0.6–1.0 | 0.060 | 0.7 | 0.5–0.9 | 0.037 |
| Abnormal blood vessels | 215/587 (37) | 337/1050 (32) | 1.2 | 0.9–1.5 | 0.063 | 1.1 | 0.8–1.3 | 0.680[a] |
| | | | | | | 1.1 | 0.9–1.4 | 0.245[b] |
| Water contact as toddler | 339/587 (68) | 611/1050 (58) | 1.5 | 1.2–1.9 | <0.001 | 1.4 | 1.1–1.8 | 0.002 |
| HIV | 86/555 (15) | 210/999 (21) | 0.7 | 0.5–0.9 | 0.008 | 0.7 | 0.5–0.9 | 0.015 |
| Red urine previously or now | 155/587 (26) | 320/1045 (31) | 0.8 | 0.6–1.0 | 0.072 | 0.8 | 0.7–1.1 | 0.138 |
| Mother present in the family | 375/587 (64) | 625/1045 (60) | 1.2 | 0.9–1.5 | 0.105 | 1.3 | 0.9–1.6 | 0.055 |
| Homogenous yellow patch | 60/587 (10) | 90/1050 (09) | 1.2 | 0.9–1.7 | 0.267 | | | |
| Has ever been pregnant | 280/587 (48) | 504/1050 (48) | 0.9 | 0.8–1.2 | 0.907 | | | |
| Heavy bleeding with clots | 176/408 (43) | 350/755 (46) | 0.9 | 0.7–1.1 | 0.292 | | | |
| Father present in the family | 136/587 (23) | 251/1045 (24) | 0.9 | 0.8–1.2 | 0.698 | | | |
| Genital itch | 361/587 (61) | 658/1046 (63) | 0.9 | 0.8–1.2 | 0.573 | | | |
| Genital burn | 243/587 (41) | 464/1046 (44) | 0.9 | 0.7–1.1 | 0.246 | | | |
| Genital sore/ulcer | 114/585 (19) | 206/1043 (20) | 0.9 | 0.8–1.3 | 0.898 | | | |
| Genital lump | 93/583 (16) | 176/1038 (17) | 0.9 | 0.7–1.2 | 0.602 | | | |
| Urge with leak | 270/587 (46) | 499/1045 (48) | 0.9 | 0.8–1.1 | 0.496 | | | |
| Urge | 202/585 (35) | 346/1044 (33) | 1.1 | 0.9–1.3 | 0.569 | | | |
| Dysuria | 297/587 (35) | 550/1044 (52) | 0.9 | 0.8–1.1 | 0.418 | | | |
| Bloody discharge | 104/583 (17) | 198/1043 (19) | 0.9 | 0.7–1.2 | 0.569 | | | |
| Watery discharge | 369/583 (63) | 638/1039 (61) | 1.1 | 0.9–1.3 | 0.452 | | | |
| Abnormal discharge smell | 260/583 (45) | 490/1042 (47) | 0.9 | 0.7–1.1 | 0.346 | | | |
| **Grainy sandy patch** | 49/587 (08) | 102/1050 (08) | 0.8 | 0.6–1.2 | 0.359 | | | |

[a]multivariable with abnormal blood vessels and current water contact

[b]multivariable with abnormal blood vessels and water contact as a toddler.

*In order to study simultaneously the impact of several variables, logistic regression analysis was applied with a 5% significance level; as a general rule variables were included in the multivariable analysis if the p-value from the crude association was less than 0.2 and if the Spearman rank correlation coefficient was below 0.7.

homogenous yellow patches), previous pregnancy, current water contact, water contact as a toddler and father present in the family were strongly associated with coming back for a follow-up investigation in both univariate and multivariable analysis. Multivariable analysis of the three FGS lesions individually (correlation) with current water contact, previous

**Table 3. Reasons for refusing follow up gynaecological examination (n = 709).**

| Reasons for denying or aborting gynaecological examination | Came for follow up | | Total (%) |
| --- | --- | --- | --- |
| | No (%) | Yes (%) | |
| Did not deny or abort | 16 (3.9) | 398 (96) | 414 (100) |
| Fear | 26 (100) | 0 (0) | 26 (100) |
| Not ready | 127 (100) | 0 (0) | 127 (100) |
| Menstruation | 24 (92) | 2 (8) | 26 (100) |
| Virgin | 14 (100) | 0 (0) | 14 (100) |
| Pregnant | 53 (100) | 0 (0) | 53 (100) |
| Other | 48 (98) | 1 (2) | 49 (100) |
| **TOTAL** | **308 (43)** | **401 (57)** | **709 (100)** |

pregnancy and water contact as a toddler had no effect on the association. A greater loss to follow-up for gynaecological investigation was observed in the Southern research clinic as compared to the Northern clinic (odds ratio = 1.9, 95% CI 1.5–2.4 and p <0.001).

On the baseline investigation day, participants were informed about genital lesions (grainy sandy patches, homogenous yellow patched and abnormal blood vessels) but urine microscopy results were not readily available as the laboratory investigation were still going on. However, only approximately 25% of the young women with lesions returned for follow-up examinations (Fig 3, bold frame). As shown in Fig 3 (hatched frame), more than 70% of learners that had genital lesions at baseline were lost to follow up for gynaecological investigation, and 23% of learners with urinary schistosomiasis were lost to follow-up.

## The effect of treatment

Multivariable analysis of the effect of treatment on baseline FGS showed that treatment only had an effect on the abnormal blood vessels (adjusted odds ratio = 2.1, 95% CI 1.1–3.9 and p = 0.018) (Table 5). Treatment was protective against urinary schistosomiasis; only 3 learners among the 55 that were treated remained positive for urinary schistosomiasis at follow up (adjusted odds ratio = 0.4, 95% CI 0.01–0.1 and p < 0.001).

Before accounting for FGS lesions at baseline (including those with and without FGS lesions at baseline), the univariate analysis showed a strong association between treatment and two FGS lesions (homogenous yellow patches and abnormal blood vessels); at multivariable analysis this association remained unchanged for the abnormal blood vessels (adjusted odds ratio = 2.3, 95% CI 1.5–3.6 and p <0.001) (Table 5) whereas it became insignificant for homogenous yellow patches.

## Discussion

The abnormal blood vessels were refractory to praziquantel treatment. Abnormal blood vessels and grainy sandy patches were previously found to be significantly associated with the presence of live worms as opposed to the homogenous yellow patches [44]. The association between praziquantel treatment and abnormal blood vessels may be an indication that abnormal blood vessels are an early stage of the grainy sandy patches and are easy to eliminate as opposed to the grainy sandy patches and homogenous yellow patches. It is important to note that different investigators define the investigation findings differently and that the investigators of gynaecological examinations were different in baseline and follow-up studies. However, the computer images were analysed by the FGS specialists afterwards for quality control.

**Table 4. Underlying factors associated with coming back for follow-up gynaecological investigations.**

| History | Came for follow-up (%) | Did not come for gynaecological investigation | Univariate analysis | | | Multivariable analysis | | |
|---|---|---|---|---|---|---|---|---|
| | | | Odds ratio | 95% Confidence Interval | p-value | Adjusted odds ratio | 95% Confidence Interval | p-value |
| Grade 8 | 5/400 (1) | 18/1863 (1) | 1 | | | 1 | | |
| Grade 9 | 24/400 (7) | 92/1863 (5) | 0.9 | 0.3–2.8 | 0.910 | 0.8 | 0.3–2.5 | 0.724 |
| Grade 10 | 120/400 (30) | 488/1863 (26) | 0.9 | 0.3–2.4 | 0.813 | 0.8 | 0.3–2.2 | 0.635 |
| Grade 11 | 214/400 (54) | 742/1863 (40) | 1.0 | 0.4–2.8 | 0.941 | 0.9 | 0.3–2.5 | 0.803 |
| Grade 12 | 37/400 (9) | 523/1863 (28) | 0.3 | 0.1–0.7 | 0.010 | 0.2 | 0.1–0.6 | 0.004 |
| Grainy sandy patch* | 56/401 (14) | 153/1892 (8) | 1.8 | 1.3–2.6 | <0.001 | 1.8 | 1.3–2.6 | <0.001 |
| Abnormal blood vessels* | 185/401 (46) | 537/1892 (28) | 2.3 | 1.7–2.7 | <0.001 | 2.2 | 1.7–2.7 | <0.001 |
| Homogenous yellow patch* | 55/401 (14) | 153/1892 (8) | 1.8 | 1.3–2.5 | <0.001 | 1.8 | 1.3–2.5 | 0.001 |
| Has ever been pregnant | 236/401 (60) | 838/1892 (44) | 1.8 | 1.4–2.2 | <0.001 | 1.8 | 1.5–2.3 | <0.001 |
| Water contact as toddler | 260/401 (65) | 1119/1892 (59) | 1.3 | 1.0–1.6 | <0.034 | 1.4 | 1.1–1.7 | 0.009 |
| Current water contact | 257/401 (64) | 1124/1892 (59) | 1.2 | 0.9–1.5 | 0.082 | 1.4 | 1.1–1.7 | 0.012 |
| Red urine previously or now | 110/398 (28) | 571/1883 (30) | 0.9 | 0.7–1.1 | 0.287 | 0.8 | 0.6–1.1 | 0.140 |
| Father present in the family | 71/398 (19) | 454/1881 (24) | 0.7 | 0.5–0.9 | 0.007 | 0.7 | 0.5–0.9 | 0.010 |
| Genital lump | 65/394 (17) | 311/1872 (17) | 0.9 | 0.7–1.3 | 0.955 | | | |
| Condom use | 130/319 (41) | 649/1604 (41) | 1.0 | 0.8–1.3 | 0.923 | | | |
| Heavy bleeding with clots | 113/259 (44) | 628/1403 (45) | 0.9 | 0.7–1.2 | 0.736 | | | |
| HIV | 83/386 (22) | 356/1794 (20) | 1.1 | 0.8–1.4 | 0.461 | | | |
| Mother present in the family | 231/399 (58) | 1136/1880 (60) | 0.9 | 0.7–1.1 | 0.349 | | | |
| Genital itch | 253/399 (63) | 1155/1884 (61) | 1.1 | 0.9–1.4 | 0.433 | | | |
| Genital burn | 179/399 (45) | 809/1884 (43) | 1.1 | 0.9–1.3 | 0.482 | | | |
| Genital sore/ulcer | 77/398 (19) | 347/1879 (19) | 1.1 | 0.8–1.4 | 0.682 | | | |
| Urge with leak (urgenic1) | 200/398 (50) | 882/1883 (47) | 1.1 | 0.9–1.4 | 0.216 | | | |
| Urge but no leak (urge1) | 137/392 (35) | 639/1881 (34) | 1.0 | 0.8–1.3 | 0.837 | | | |
| Dysuria | 217/398 (55) | 982/1882 (52) | 1.1 | 0.9–1.4 | 0.395 | | | |
| Bloody discharge | 70/397 (18) | 356/1875 (19) | 0.9 | 0.7–1.2 | 0.530 | | | |
| Watery discharge | 252/396 (64) | 1168/1872 (62) | 1.1 | 0.8–1.3 | 0.642 | | | |
| Abnormal discharge smell | 179/399 (45) | 878/1869 (47) | 0.9 | 0.7–1.1 | 0.442 | | | |
| Urinary schistosomiasis | 82/367 (22) | 322/1579 (20) | 1.1 | 0.9–1.5 | 0.407 | | | |

*Were analysed separately because some participants were affected by more than one lesion and there may be causal relationship between the three FGS lesions.

South Africa is not implementing mass drug administration [45]; the remaining gynaecological symptoms among the treated were high and may be an indication that for effectiveness, several rounds of treatment are required and should be started at an earlier age. A similar study in Zimbabwe, which investigated the effect of praziquantel on FGS, reported a result consistent to ours where a standard single-dose of praziquantel did not have effect on the

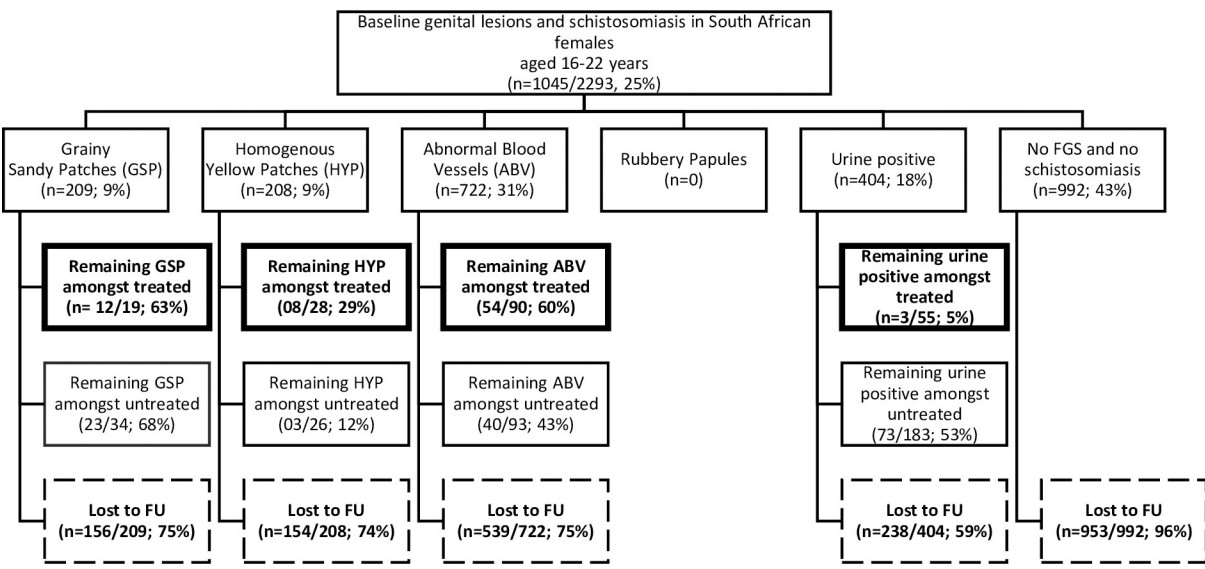

**Fig 3. Follow-up amongst adolescent girls and young women investigated for Female Genital Schistosomiasis in Kwa-Zulu-Natal.**
Hatched frame: lost to follow-up for gynaecological investigation, Bold frame: Remaining findings at follow-up gynaecological investigation.

inflammatory lesions of FGS that had formed and developed over a few years [13]. The timing for follow up investigations may have been too soon for some learners to determine the true effect of treatment (the median time period between praziquantel (40mg/kg) treatment and reinvestigation was 5 months (range 5–14 months).

Treatment with praziquantel is the only recommended effective way of killing adult helminths and mature live eggs, but has no effect on the calcified ova; early treatment is therefore crucial before the ova are deposited in the tissue and cause damage [6,46]. This may possibly explain the remaining gynaecological symptoms among the treated, including those who did not have the lesions at baseline. A retrospective study found that treatment at an early age seems to prevent gynaecologic morbidity [13].

Lack of a regular mass treatment programmes in South Africa, repeated exposure to risky water, and re-infections could have led to continuous FGS lesions even after a single dose of praziquantel. Further investigations and follow up studies are therefore needed in countries that are implementing mass drug administration and areas that are not implementing mass

**Table 5. Effect of treatment—remaining FGS lesions among the treated and untreated.**

| FGS lesions | Treated (%) | Untreated (%) | Univariate analysis | | | Multivariable analysis | | |
|---|---|---|---|---|---|---|---|---|
| | | | OR | 95% CI | p-value | AOR | 95% CI | p-value |
| **Effect of treatment among those with FGS a baseline** | | | | | | | | |
| Grainy sandy patches | 12/19 (63) | 23/34 (68) | 0.8 | 0.3–2.7 | 0.741 | 0.5 | 1.1–2.1 | 0.369 |
| Homogenous yellow patches | 08/28 (29) | 03/26 (12) | 3.1 | 0.7–3.2 | 0.120 | 6.7 | 0.9–48 | 0.058 |
| Abnormal blood vessels | 54/90 (60) | 40/93 (43) | 1.9 | 1.1–3.6 | 0.022 | 2.1 | 1.1–3.9 | 0.018 |
| Urinary schistosomiasis | 3/55 (5) | 73/183 (53) | 0.5 | 0.02–0.2 | <0.001 | 0.4 | 0.01–0.1 | <0.001 |
| **Overall FGS rate at follow up (after treatment and including FGS among those who did not have FGS at baseline)** | | | | | | | | |
| Grainy sandy patches | 25/167 (15) | 35/229 (15) | 0.9 | 0.6–1.7 | 0.931 | 0.9 | 0.5–1.7 | 0.903 |
| Homogenous yellow patches | 24/167 (14) | 17/229 (07) | 2.1 | 1.1–4.0 | 0.025 | 1.8 | 0.9–3.5 | 0.096 |
| Abnormal blood vessels | 81/167 (49) | 66/229 (29) | 2.3 | 1.5–3.5 | <0.001 | 2.3 | 1.5–3.6 | <0.001 |
| Urinary schistosomiasis | 07/351 (02) | 91/681 (13) | 0.1 | 0.1–0.3 | <0.001 | 0.1 | 0.1–0.3 | <0.001 |

drug administration. These studies will help with information to fully understand the differences between the FGS lesions and whether praziquantel treatment can eliminate these lesions, including the right timing to possibly eliminate the lesions. Currently, only one study has investigated the impact of treatment on gynaecological lesions and found no significant change in the adult sandy patches and contact bleeding over a 12-month period, even though urinary egg excretion ceased [43]. Another study found that schistosomiasis PCR remained positive in the genitals after treatment [42]. The lack of information and complexity in conducting research on how long it takes for the FGS lesions to manifest following infection made it difficult to determine whether the FGS lesions diagnosed at follow-up were due to schistosomiasis re-infection or were progressive lesions that had not yet manifested and therefore diagnosed at baseline. Lack of a regular mass treatment programme in South Africa, repeated exposure to risky water, and re-infections could have led to continuous FGS lesions even after a single dose of praziquantel.

Identification of HIV, current water contact, water contact as a toddler and urinary schistosomiasis as factors that influenced learners to participate in mass treatment may be a sign of some awareness of risk factors of schistosomiasis, i.e. schistosomiasis as a risk factor for HIV, and water contact as a risk for schistosomiasis. The pre-treatment health education may have played a role in influencing these participants to accept the treatment. Learners who felt they do not have a problem with these identified factors may have perceived it unnecessary for them to get treated. Furthermore, the three FGS lesions (grainy sandy patches, abnormal blood vessels and homogenous yellow patches), previous pregnancy, current water contact, water contact as a toddler and father present in the family were strongly associated with returning for follow-up investigation. It is anticipated that the investigators at the Northern clinic had gained more experience in both health education and community engagement as opposed to when they conducted investigations at the Southern clinic (the 1st investigation site) where there was a greater loss to follow up. Therefore, active ongoing community engagement targeting both learners, parents and the entire community at risk is crucial in order to empower the community with the full knowledge on prevention and control that will enable them to make informed decisions.

South Africa is endemic for schistosomiasis with some focal areas having high and moderate infections [44], and FGS studies among the young population may help identify targeted interventions. Study findings revealed that more than 20% of the adolescent women in KwaZulu-Natal province had three well-known genital mucosal manifestations of FGS [30]; in Limpopo Province, FGS accounted for 87.6% of the female cases in a study that described the pathology of biopsy diagnosed schistosomiasis [47]. However, treatment in South Africa is still case-based and, as in many other countries, most community members do not seek early treatment or do not seek treatment at all [17,27,48]. Therefore, those infected and those that complicate to FGS remain infected or with FGS complication; some individuals are misdiagnosed as sexually transmitted infections (STIs) or cervical cancer [2,5,20–27]. In this study, loss to follow up was a major challenge. Involving community health workers in FGS studies may assist with managing those who are lost to follow up because it would be easier for them to maintain contact and follow up with the learners in their catchment area, especially those found to have FGS symptoms.

It is important to note that gynaecological examination was not possible among the early and middle adolescent group due to virginity, fear, pregnancy, and privacy and other related issues. Early adolescence and some middle adolescence may only be targeted for prevention and only examined at the late adolescence stage to check if they developed any FGS after repeated treatment. Observational studies in young adolescents and adults are critical in determining the effect of early treatment on schistosomiasis and FGS. It is similarly important to determine the burden of infertility among those living in endemic areas and determine the

association with FGS. Therefore, it is important for countries such as South Africa to establish mass drug administration programmes (targeting communities at risk, prioritizing enrolled and unenrolled school age children, as well as adolescent girls and women of reproductive age) in order to prevent sustained infections that can lead to FGS.

The high loss to follow up is worrying because more knowledge is required through research to understand FGS pathogenesis and improve the knowledge of clinicians. For instance, it is important to note that the high loss to follow-up during mass treatment in this study contributed to small sample size to determine the effect of treatment on FGS. Currently, the lack of knowledge of FGS among most clinicians and affected communities is concerning due to the reported high prevalence of genital lesions [49]. It has been reported that up to 75% of all women and girls infected with urinary *S. haematobium* have lesions in the uterus, cervix, vagina or vulva [4]. Many women have been reported to have genital schistosomiasis without urinary excretion of *S. haematobium* eggs [2,4,13,50,51]. In sub-Saharan Africa, it is estimated that 56 million women have FGS [2].

In addition to the identified unintended consequences due to inclusion criteria and research actions reported under results, the high loss to follow-up and low treatment coverage in this study could also be attributed to factors that were reported by another study in South Africa in Ugu Distict of KwaZulu-Natal Province. This Ugu District study was conducted among grade 10–12 learners, teachers, community health workers and traditional healers [16,17]. Factors that contributed to low coverage in this Ugu District study were reported to be: older age group, lack of knowledge, attending a large school, parental control and a closer teacher follow-up in younger children and in small schools, misconceptions that schistosomiasis is a self-healing disease and symptoms confused with sexually transmitted infections, the chronicity of the disease is not known to the general population, teasing and stigma, schistosomiasis-related absenteeism that may reach 30% on some days, tablets must be distributed by health professionals in South Africa, and schools reluctant to provide more than one day for treatment in order to minimise disturbance [15,17,27].

The lower loss to follow up at the North of Durban clinic may be attributed to the experience gained while operation at the first clinic (South of Durban). A praziquantel mass treatment programme was implemented in South Africa, in Ugu District of KwaZulu-Natal Province between 1998 and 2001 among grade 10–12 learners, teachers, community health workers and traditional healers, and only reached 44% of the learners instead of the WHO and the South African National Department of Health recommended target of 75% coverage [27]. This shows a need to embark on large scale awareness and advocacy campaign in schools and communities, coordinated by the Department of Health and in collaboration with the Department of Basic Education, non-governmental organisations and community leaders, before implementing FGS studies among young people and mass treatment administration, to improve participation in research that will inform control measures.

Study limitations include the small sample size to determine the effect of treatment due to low treatment coverage and loss to follow up. Another limitation was that treatment was not done immediately after baseline, and therefore many learners were not treated between baseline and follow up either due to factors described above by Lothe *et al* in the qualitative studies in Ugu District, KwaZulu-Natal Province [16,17]. As a result, some learners that were not treated between baseline and follow up were only treated during follow-up. Some of the learners under 16 years might already have had chronic lesions, and inclusion of the Grade 12 learners could have contributed to loss to follow up because of those who passed matric and moved to tertiary institutions the following year.

In summary, it is critical to conduct vigorous ongoing community engagements for awareness and buy-in prior to mass treatment and gynaecological investigations in order to improve

community participation and increase sample size. The effect of treatment was investigated among a small sample size and there was loss to follow-up which could have been prevented by treatment immediately after gynaecological investigation at the clinic at baseline. It is therefore difficult to draw firm conclusions about the effect of treatment on FGS lesions. In addition to the vigorous community engagement, future studies should prioritize treatment immediately after baseline investigation as another strategy for improving sample size.

## Acknowledgments

We are appreciative of the support from Roy Manyaira, Silindile Gagai and other staff at BRIGHT Research in KwaZulu-Natal, South Africa. We thank all the girls and young women who participated in the study.

## Author Contributions

**Conceptualization:** Takalani Girly Nemungadi, Elisabeth Kleppa, Hashini Nilushika Galappaththi-Arachchige, Pavitra Pillay, Svein Gunnar Gundersen, Birgitte Jyding Vennervald, Patricia Doris Ndhlovu, Myra Taylor, Eyrun Floerecke Kjetland.

**Data curation:** Takalani Girly Nemungadi.

**Formal analysis:** Takalani Girly Nemungadi.

**Funding acquisition:** Eyrun Floerecke Kjetland.

**Investigation:** Takalani Girly Nemungadi, Elisabeth Kleppa, Hashini Nilushika Galappaththi-Arachchige, Patricia Doris Ndhlovu, Eyrun Floerecke Kjetland.

**Methodology:** Takalani Girly Nemungadi, Eyrun Floerecke Kjetland.

**Project administration:** Eyrun Floerecke Kjetland.

**Supervision:** Saloshni Naidoo, Eyrun Floerecke Kjetland.

**Validation:** Takalani Girly Nemungadi, Eyrun Floerecke Kjetland.

**Writing – original draft:** Takalani Girly Nemungadi.

**Writing – review & editing:** Takalani Girly Nemungadi, Elisabeth Kleppa, Hashini Nilushika Galappaththi-Arachchige, Pavitra Pillay, Svein Gunnar Gundersen, Birgitte Jyding Vennervald, Patricia Doris Ndhlovu, Myra Taylor, Saloshni Naidoo, Eyrun Floerecke Kjetland.

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
