## [Decision Letter · Decision Letter 0]

18 Dec 2023

Dear Ms Nemungadi,

Thank you very much for submitting your manuscript "PREDICTORS FOR PARTICIPATION IN MASS-TREATMENT AND FEMALE GENITAL SCHISTOSOMIASIS RE-INVESTIGATION, AND THE EFFECT OF PRAZIQUANTEL TREATMENT IN SOUTH AFRICAN ADOLESCENTS" for consideration at PLOS Neglected Tropical Diseases. As with all papers reviewed by the journal, your manuscript was reviewed by members of the editorial board and by several independent reviewers. In light of the reviews (below this email), we would like to invite the resubmission of a significantly-revised version that takes into account the reviewers' comments. 

We cannot make any decision about publication until we have seen the revised manuscript and your response to the reviewers' comments. Your revised manuscript is also likely to be sent to reviewers for further evaluation.

Sincerely,

Luc E. Coffeng, MD PhD

Guest Editor

Eva Clark

Section Editor

Please also address these minor point that Reviewer 2 provided in a separate word document:

Overall, the paper is well written and addresses an important topic related to reproductive health of women and adolescent girls living in schistosomiasis endemic areas. I have only few observations:-

Line 145: use participants instead of study subjects

Line 138-150- remove the bullets- put them in sentences

Reviewer's Responses to Questions

**Key Review Criteria Required for Acceptance?**

**Methods**

-Are the objectives of the study clearly articulated with a clear testable hypothesis stated?

-Is the study design appropriate to address the stated objectives?

-Is the population clearly described and appropriate for the hypothesis being tested?

-Is the sample size sufficient to ensure adequate power to address the hypothesis being tested?

-Were correct statistical analysis used to support conclusions?

-Are there concerns about ethical or regulatory requirements being met?

Reviewer #1: **Major Revision**

The study's objectives are clearly defined, and the chosen population and study site are deemed appropriate, despite encountering low participation. This acknowledged limitation is discussed in the relevant section of the study. However, there are notable gaps and inconsistencies in the methodology that require clarification and improvement. The following considerations and suggestions aim to enhance the overall clarity of the methodology and study design:

1) Clarification on Data Collection and Analysis:

Please provide clarification on whether the data collection, conducted between 2011 and 2013, was retrospective. Additionally, specify when the analysis was conducted, especially considering the intent to publish in 2023. Discuss the implications of this temporal aspect in the limitation chapter.

2)FGS Lesion Detection and Data Processing:

It remains unclear from the methodological description how FGS lesions were detected, including details on data collection and processing. Specify the instruments used, the individuals performing the examination, and their level of training. Discuss whether images were reviewed by independent specialists and, if not, provide the rationale. Address the implications of these aspects in the results in the limitation chapter.

3) Clarification on Necessary Sample Size (Table 1, Row 3):

Elaborate on what the necessary sample size referred to in Table 1, Row 3 represents. Provide details on how this figure was determined and its significance to the study.

4) Reference for Line 190 and Rationale for Sample Size (Line 190):

Include a reference for line 190, where the need to study 318 FGS patients and 954 without FGS is mentioned. Provide a more explicit explanation of the rationale behind these specific numbers and how they were derived from the probabilities mentioned in line 190.

5) Clarification on Venn Diagram Reference (Line 203):

Clarify whether the Venn diagram referenced in line 203 is indeed referring to Figure 2 on page 12. Clearly state this connection in the text to avoid any confusion.

These adjustments will contribute to a more comprehensive and transparent presentation of the study methodology, addressing specific concerns and improving the overall quality of the research.

Reviewer #2: -The objectives are very clear

-The study design is appropriate. 

-The sample size is well describe with hypothesis testing with adequate power

-The analysis plan is very clear. 

-All ethical procedures were followed except that treatment were offered after a long time of waitinf ro results. The best information is that everyone receive treatment

**Results**

-Does the analysis presented match the analysis plan?

-Are the results clearly and completely presented?

-Are the figures (Tables, Images) of sufficient quality for clarity?

Reviewer #1: **Major Revision**

The analysis presented aligns with the analysis plan; however, due to certain methodological gaps, some of the results lack clarity. To ensure consistency in data presentation, here are specific considerations and recommendations for the author to enhance the results section:

 1) Clarification on "Signs of Urinary Schistosomiasis at Baseline" (Line 211):

 In the methodology section, please provide clarification on what is meant by "signs of urinary schistosomiasis at baseline" (line 211). Specify whether this data was collected during the questionnaire phase or the clinical examination phase.

2) Add Year to Line 227:

In line 227, please add the year for reference.

3) Table 2 and Table 3 - Confidence Interval and P-Value (Grade 8 - Univariate Analysis):

In Table 2 and Table 3, include Confidence Intervals and P-values for Row 2 (Grade 8) in the univariate analysis. Additionally, address missing data in the multivariable analysis columns. Consider reporting this in a separate table or marking the missing data/analysis (e.g., N/A) in the rows.

4) Replace "To Account for a Causal Relationship" with "Correlation" (Line 16, Page 14):

On line 16, page 14, replace "to account for a causal relationship" with "correlation" for clarity.

5) Table 3 - Consistency in Data Presentation:

In Table 3, ensure consistency in how data is presented. Use a uniform format such as "n=208; 9%" throughout, avoiding variations like "(54/90; 60%)" for the sake of clarity and coherence

Reviewer #2: The results match the objectives and the analysis plan

-The results are clear and complete

-The tables are well presented

**Conclusions**

-Are the conclusions supported by the data presented?

-Are the limitations of analysis clearly described?

-Do the authors discuss how these data can be helpful to advance our understanding of the topic under study?

-Is public health relevance addressed?

Reviewer #1: **Major Revision**

The discussion and conclusion chapters require a thorough copy-editor review. The discussion section establishes intriguing connections with past studies but lacks organizational coherence, making it challenging to deliver a clear message. Some sentences are difficult to grasp, hindering the communication of the main implications of the study. Here are specific considerations and remarks to enhance clarity in the discussion and conslusion sections:

1) Rewrite Sentence for Clarity (Line 40, Page 18):

"It is important to note that different investigators define the investigation findings differently, and that the investigators of gynecological examinations were different in baseline and follow-up studies."

 Clarification: Does this refer to the data collection of visual inspections for the presentation of lesions? It is not clear in the methodology how this comes about, and this discussion sentence contributes to the confusion.

2) Clarify Phrasing (Line 47, Page 18):

Confusing phrasing: "This supports findings in Zimbabwe."

 Clarity: "A similar study in Zimbabwe, which investigated [add specific details], reported a result consistent with ours. Where a standard (…)"

3) Rewrite Complex Sentence (Lines 78-81, Page 19):

 "Considering the interplay of various factors influencing participation and the contextual dynamics, the observed outcomes may be subject to nuanced interpretations."

 Simplification: Rewrite the sentence for better understanding.

4) Specify MDA Target Audience (Discussion Chapter):

Add to the discussion chapter details about who the Mass Drug Administration (MDA) should target. Considering the emerging consensus that it should not be limited to school-age children, there is a call to include women and girls in reproductive age, as well as efforts to reach the population that did not attend school.

5) Address Methodological Limitations (Methodological Review Reference):

Acknowledge and address the methodological limitations referenced in the methodological review. Provide insights into how these limitations may have impacted the study and suggest potential avenues for future research to mitigate these constraints.

Reviewer #2: COnclusion is supported by the data though need to be written to focus on the results not an opinion/a recommendation. What is given is a recommendation not a conclusion

**Editorial and Data Presentation Modifications?**

Reviewer #1: **Major Revision**

To enhance the impact of the study, it is recommended to refine the article's structure, address methodological gaps, and ensure consistency in data presentation. The authors should also focus on improving readability. 

Here are further specific remarks and suggestions for the Abstract, Author Summary, and Introduction:

A) ABSTRACT:

 Methods (Lines 37-38):

 1)Specify the total sample size (n) for sample recruitment.

 2) Clarify the statement "a few in September" by providing the actual number and explaining the reasons for the decrease in numbers during September.

 3) Year of the Study (Line 40):

 Specify the year of the study to provide a clear timeframe for readers.

 Results Section:

 4) Organize the results under distinct themes for better clarity:

 Prevalence of Schistosomiasis

 Participation Rates

 Loss to Follow-up among FGS Lesions

 Follow-up Findings

 Factors Influencing Treatment and Follow-up Participation (Mass Treatment & Follow-up Gynaecological Investigation)

 Challenges in Sample Size for Follow-up Analysis

 Multivariable Analysis results

 5) Clarify Loss to Follow-up (Line 48):

 Clarify if the 70% loss refers to the total sample size of 2293 or specifically to the 1045 learners with FGS lesions.

 Provide the total sample size for the follow-up group and clarify the proportion of grade 12 students in the loss.

 6) Consistent Data Presentation:

 Present data consistently, using the same format throughout (e.g., "12 (of 19)" and "3 out of 5"). Remove the term "huge" from line 54 for a more objective description.

 7) Improved Conclusion:

 Revise the conclusion for clarity and impact:

 "Despite challenges in sample size and significant loss to follow-up among those with baseline FGS, limiting the ability to fully understand the treatment's effect, multivariable analysis demonstrated a significant treatment effect on abnormal blood vessels."

B) AUTHOR SUMMARY:

 8)Line 76:

 Replace "diagnosed as suffering from diseases such as" with "misdiagnosed with."

 9) Line 87:

 Remove the term "huge" for more measured language.

C) INTRODUCTION:

 10) Reference to WHO Policy Brief (Last Paragraph):

 Add a brief sentence referring to the recently published WHO policy brief titled "Deworming adolescent girls and women of reproductive age. Policy brief."

Reviewer #2: None

**Summary and General Comments**

Reviewer #1: This study is significant within the context of Female Genital Schistosomiasis (FGS). The manuscript aims to investigate FGS in young women (16–23 years) attending rural high schools in South Africa. It explores predictors for accepting anti-schistosomal treatment and/or gynaecological reinvestigation, examining the effects of mass-treatment (praziquantel) on FGS clinical manifestations in adolescents. The research encompasses baseline gynaecological investigations, mass-treatment, and reinvestigation after a median of 9 months.

Key findings underscore a high prevalence of FGS signs, coupled with low participation in mass treatment and reinvestigations, leading to substantial loss to follow-up. Multivariable analysis identifies a treatment effect specifically on abnormal blood vessels. Despite the relevance of the content, the overall structure of the article needs improvement for better clarity and logic.

There are notable gaps in the methodology, particularly in the processing of data collection, and inconsistencies in data presentation. It is crucial to acknowledge earlier in the manuscript that the data were collected more than 10 years ago, specifying the implications of this temporal gap. Additionally, the timing of data analysis remains unclear.

The discussion section establishes intriguing connections with past studies but lacks organizational coherence, making the delivery of a clear message challenging. To enhance the study's impact, it is recommended to refine the article's structure, address methodological gaps, and ensure consistency in data presentation. Furthermore, explicit acknowledgment of the data's age and clarity on the timing of data analysis will contribute to a more robust interpretation

Reviewer #2: Overall, the paper gives the way fowards for schistosomaisis control in South Africa, specifically, the paper addresses the issue of FGS, a hot topic with so many questions in term of diagnosis and treatment outcome

PLOS authors have the option to publish the peer review history of their article (what does this mean?). If published, this will include your full peer review and any attached files.

Reviewer #1: No

Reviewer #2: Yes: Humphrey D. Mazigo
---

## [Decision Letter · Decision Letter 1]

12 Mar 2024

Dear Ms Nemungadi,

We are pleased to inform you that your manuscript 'PREDICTORS FOR PARTICIPATION IN MASS-TREATMENT AND FEMALE GENITAL SCHISTOSOMIASIS RE-INVESTIGATION, AND THE EFFECT OF PRAZIQUANTEL TREATMENT IN SOUTH AFRICAN ADOLESCENTS' has been provisionally accepted for publication in PLOS Neglected Tropical Diseases.

Best regards,

Eva Clark, M.D., Ph.D.

Section Editor

Eva Clark

Section Editor

Reviewer's Responses to Questions

**Key Review Criteria Required for Acceptance?**

**Methods**

-Are the objectives of the study clearly articulated with a clear testable hypothesis stated?

-Is the study design appropriate to address the stated objectives?

-Is the population clearly described and appropriate for the hypothesis being tested?

-Is the sample size sufficient to ensure adequate power to address the hypothesis being tested?

-Were correct statistical analysis used to support conclusions?

-Are there concerns about ethical or regulatory requirements being met?

Reviewer #1: *Accept*

The authors addressed changes required for the methods section. The objectives are clearly stated, the study designed is presented in a organized way.

**Results**

-Does the analysis presented match the analysis plan?

-Are the results clearly and completely presented?

-Are the figures (Tables, Images) of sufficient quality for clarity?

Reviewer #1: *Accept*

The authors addressed changes required for the results section. The results are clearly stated, the tables are presented in a organized way.

**Conclusions**

-Are the conclusions supported by the data presented?

-Are the limitations of analysis clearly described?

-Do the authors discuss how these data can be helpful to advance our understanding of the topic under study?

-Is public health relevance addressed?

Reviewer #1: *Accept*

The authors addressed changes required for the discussion and conclusion section. The study limitations are well described.

**Editorial and Data Presentation Modifications?**

Reviewer #1: *Accept*

No further changes are needed. The manuscript read well with incredible contributions for the FGS research field. Well done with the revision.

**Summary and General Comments**

Reviewer #1: (No Response)

PLOS authors have the option to publish the peer review history of their article (what does this mean?). If published, this will include your full peer review and any attached files.

Reviewer #1: No

---

## [Editor Report · Acceptance letter]

21 Mar 2024

Dear Ms Nemungadi,

We are delighted to inform you that your manuscript, "PREDICTORS FOR PARTICIPATION IN MASS-TREATMENT AND FEMALE GENITAL SCHISTOSOMIASIS RE-INVESTIGATION, AND THE EFFECT OF PRAZIQUANTEL TREATMENT IN SOUTH AFRICAN ADOLESCENTS," has been formally accepted for publication in PLOS Neglected Tropical Diseases.

Best regards,

Shaden Kamhawi

co-Editor-in-Chief

Paul Brindley

co-Editor-in-Chief
